# A multi-sample approach increases the accuracy of transcript assembly

Li Song[1,2,5], Sarven Sabunciyan[3], Guangyu Yang[1,2] & Liliana Florea [iD] [1,2,4]*

Transcript assembly from RNA-seq reads is a critical step in gene expression and subsequent functional analyses. Here we present PsiCLASS, an accurate and efficient transcript assembler based on an approach that simultaneously analyzes multiple RNA-seq samples. PsiCLASS combines mixture statistical models for exonic feature selection across multiple samples with splice graph based dynamic programming algorithms and a weighted voting scheme for transcript selection. PsiCLASS achieves significantly better sensitivity-precision tradeoff, and renders precision up to 2-3 fold higher than the StringTie system and Scallop plus TACO, the two best current approaches. PsiCLASS is efficient and scalable, assembling 667 GEUVADIS samples in 9 h, and has robust accuracy with large numbers of samples.

[1] McKusick-Nathans Institute of Genetic Medicine, Johns Hopkins School of Medicine, Baltimore, MD, USA. [2] Department of Computer Science, Johns Hopkins University, Baltimore, MD, USA. [3] Department of Pediatrics, Johns Hopkins School of Medicine, Baltimore, MD, USA. [4] Department of Medicine, Johns Hopkins School of Medicine, Baltimore, MD, USA. [5] Present address: Department of Data Sciences, Dana Farber Cancer Institute, Boston, MA, USA. *email: florea@jhu.edu

RNA sequencing (RNA-seq) has become the de facto standard in surveying the transcriptome of a cell, organism, or species, to determine the expressed genes and transcripts and their expression levels, and to enable differential and functional analyses[1,2]. A crucial step in virtually all RNA-seq data analyses is assembling the reads into full-length transcripts. The accuracy of transcript reconstruction is critical for quantification, detection, and characterization of alternative splice variants, and the identification of differences in gene expression and splicing patterns between tissues, developmental stages, and physiological or disease states.

Virtually all transcriptomic studies involve multiple samples. The current paradigm is to assemble the reads in each sample, then merge the partial transcripts (transfrags) across all samples to create a unified set of meta-annotations[3], which is used as reference for downstream quantification and differential analyses. Most single-sample assemblers including Cufflinks, isoCEM, Scripture, Traph, CLASS, iReckon, CIDANE, FlipFlop, CLASS2, StringTie, Scallop, and TransComb[4–15] build a graph structure from read alignments on the genome, then traverse the graph to select an optimized set of transcripts, represented as paths. Recent second-generation transcript assembly methods including StringTie, CLASS2, and Scallop have taken great strides toward increasing the accuracy and efficiency of assembly at single-sample level, and meta-assemblers such as StringTie(ST)-merge and TACO[3] have led to more robust collections of meta-annotations. Despite these efforts precision remains low, with <40% of the predicted transcripts in a single-sample and <30% of transcripts in meta-annotations representing complete and accurate reconstructions[3,16]. Concomitantly, a handful of efforts have focused on designing methods that simultaneously assemble transcripts across multiple RNA-seq samples, notably CLIIQ, ISP, MiTie, and FlipFlop[11,17–19]. However, none of these programs has comparable performance or is equipped to handle more than a limited number of samples. (See Supplementary Methods for a more in-depth review of related work.)

We present PsiCLASS, based on an approach that simultaneously analyzes multiple RNA-seq samples, which achieves significantly higher precision at sensitivity comparable to the best current approaches, and significantly higher overall accuracy in its default setting. PsiCLASS differs from traditional assemblers in two ways. First, it is a combined assembler, reporting a set of transcripts for each sample, and meta-assembler[3], producing a set of meta-annotations obtained by combining the individual samples' outputs. Second, unlike traditional single-sample assemblers, it uses information from all samples to produce transcripts for individual samples and for the unified annotation set. PsiCLASS starts by selecting a set of high-confidence introns and subexons at each locus, using statistical models of introns and intronic read levels. In doing so, it first generates a set of features (subexons, introns) for each sample, and then corroborates information across all samples to select a high-confidence subset. Next, PsiCLASS builds a unified subexon splice graph that is used within a dynamic programming optimization procedure to select a representative set of transcripts in each sample (see "Methods" and Supplementary Fig. 1 for details). Lastly, PsiCLASS extracts a subset of meta-annotations from the aggregated transcript sets by voting. Salient features of PsiCLASS include: (i) an algorithm that assembles reads from multiple samples simultaneously, taking advantage of their redundancy and complementarity; (ii) significantly higher precision at meta-assembly level, up to 2–3 fold, over StringTie with ST-merge and Scallop with TACO, and high precision (>50%) overall; (iii) 22–140% increase in precision at meta-assembly level over StringTie with ST-merge, and 35–89% over Scallop with TACO, when matched to their sensitivity levels; (iv) higher overall performance in combined sensitivity and precision at the individual sample level; (v) improved consistency among the individual samples' annotations and meta-annotations, and robustness with different meta-assemblers; and (vi) high efficiency and scalability, taking only 9 h to assemble 667 GEUVADIS samples, and robust accuracy as the number of samples increases. Overall, PsiCLASS is highly efficient and overcomes limitations in the existing methods, showing significantly higher accuracy, in particular precision, on data sets with a handful to hundreds of samples, thus providing an efficient method for RNA-seq data analysis.

## Results

**Performance evaluation on simulated data.** We compared PsiCLASS with the best current approaches, namely StringTie and Scallop at the individual sample level, and the combinations of StringTie with ST-merge and Scallop with TACO, at the meta-assembly level. We also included FlipFlop, which was the only other competitive multi-assembler; because of excessive run times, however, FlipFlop was not feasible for our tests on real data sets. (Other combinations are shown in Supplementary Fig. 2). We first applied the methods to 25 RNA-seq samples simulated with Polyester[20], where reads were aligned with two methods, Hisat2[21] and STAR[22]. Performance was slightly better for all programs when reads were aligned with Hisat2 (Supplementary Fig. 3), therefore we chose this alignment method for the rest of the analyses.

On the simulated data, PsiCLASS with default voting achieved 71.4% precision, which is 15.0% higher than the StringTie system, and 28.3% and 21.6% higher than Scallop and FlipFlop, respectively, combined with TACO, whereas sensitivity for all programs was roughly 50% (Fig. 1a). Even at the individual sample level, PsiCLASS had both the highest precision and the highest sensitivity: 73.8% precision on average, compared with 70.8% for StringTie, 62.9% for Scallop, and 50.7% for FlipFlop, and 53.4% sensitivity compared with 41.7% for StringTie, 46.2% for Scallop, and 36.1% for FlipFlop. Precision values for StringTie, Scallop, and FlipFlop, but to a lesser extent for PsiCLASS, dropped significantly after aggregation, hence PsiCLASS produces more consistent sets of transcripts between individual samples and the set of meta-annotations.

We further investigated the performance of methods based on the transcripts' expression levels (Fig. 1b). Simulated transcripts were divided into low (463 transcripts; Fragments Per Kilobase (FPK) < 30), medium (658 transcripts; $30 \leq FPK < 500$) and high (322 transcripts; $FPK \geq 500$) according to the predefined expression levels. PsiCLASS with voting reconstructs the largest fraction of high-expressed transcripts, 82.4%, and all three programs recover ~60% of the medium-expressed ones. FlipFlop has by far the highest sensitivity in detecting low expression features, 41.0%, compared with 10–20% for the other programs. Note that, because a reconstructed transcript's expression level may fall in another class than the predefined one, precision cannot be rigorously evaluated.

Similarly, to assess the programs' performance based on the gene's splicing complexity, we divided genes into three categories by the number of their alternatively spliced transcripts: low (1 transcript per gene; 680 genes, 680 transcripts), medium (2 transcripts per gene; 166 genes, 332 transcripts), and high (3 or more transcripts per gene; 106 genes, 431 transcripts). While, StringTie showed the best overall performance on the single-transcript genes, PsiCLASS had the highest sensitivity and precision on the medium and high complexity classes, in particular a 16.8–24.5% gain in sensitivity and 21.0–31.8% in precision on highly expressed transcripts (Supplementary Fig. 4).

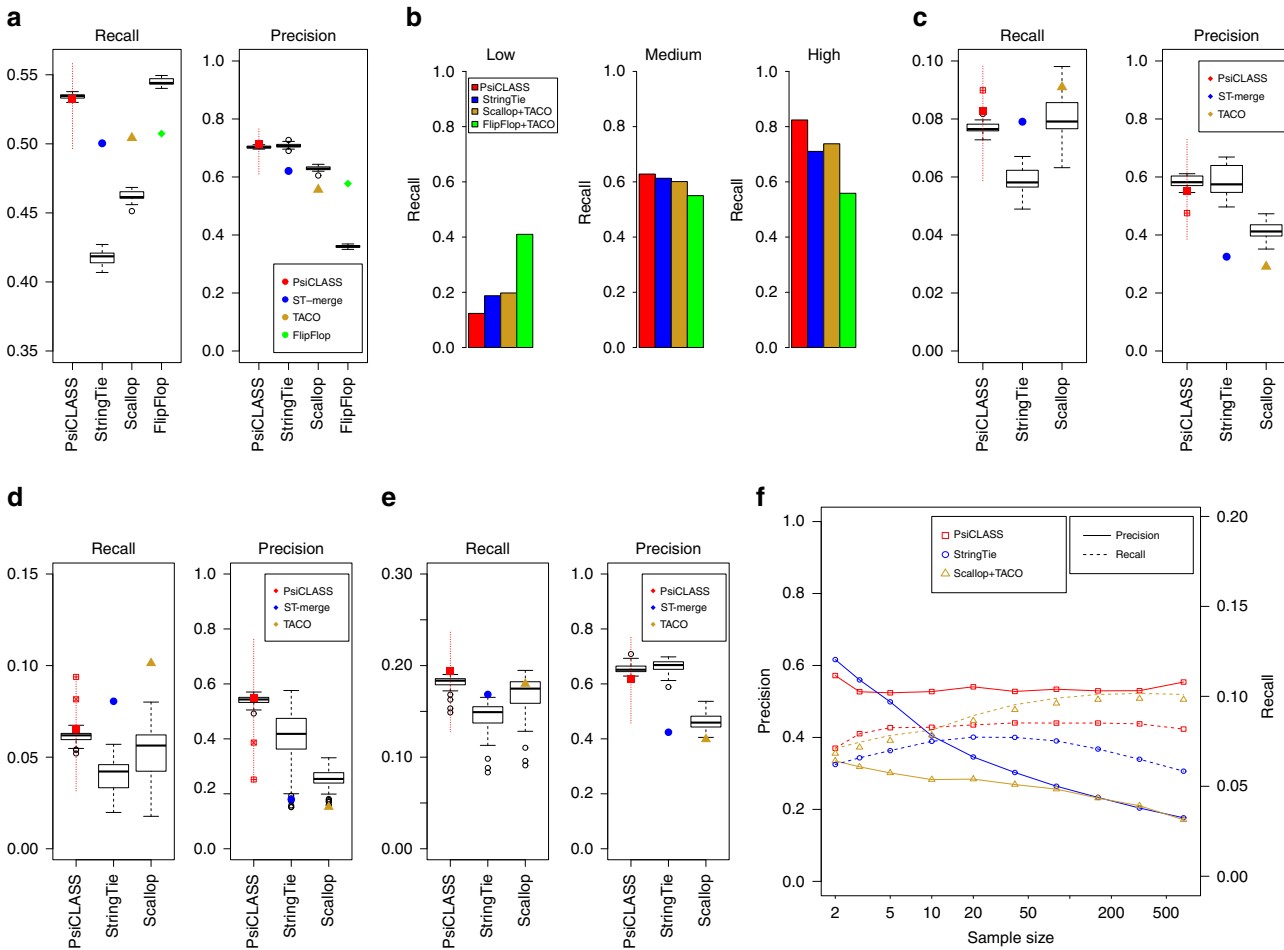

**Fig. 1** Performance evaluation of methods on simulated and real data: **a** 25 simulated RNA-seq sets, all genes; **b** 25 simulated sets, genes grouped by abundance; **c** 25 GEUVADIS samples (polyadenylated RNA); **d** 73 liver RNA-seq samples (rRNA-depleted total RNA); **e** 44 hippocampus samples from healthy and epileptic mice; and **f** subsets of 1, 2, 3, 5, 10, 20, 40, 80, 160, and 320, and the full set of 667 GEUVADIS samples. In **a**, **b**–**e**, sensitivity (recall) and precision values for PsiCLASS, StringTie, and Scallop at the level of individual samples are shown in boxed plots, and meta-annotations resulted from aggregation (with PsiCLASS voting, ST-merge, and TACO) are shown with colored shapes. Boxplots were generated with R using default options, namely, the box extends between the lower and upper quartiles, horizontal lines mark the median, and whiskers are at the 1.5 × interquartile point above the upper quartile and bellow the lower quartile. Lastly, additional symbols mark sensitivity and precision values for PsiCLASS when tuned to match or approach the sensitivity of its competitors, along the range of cutoff values 0–16, shown with dotted red lines. Source data are provided as a Source Data file

**Performance evaluation on real RNA-seq data collections**. We next assessed the performance on two representative RNA-seq data sets, generated with two different library preparation protocols: 25 randomly selected sets from polyA-selected lymphoblastoid samples from the GEUVADIS population variation project, and 73 rRNA-depleted total RNA libraries from postmortem human liver samples with funding from the Stanley Medical Research Institute. At the meta-assembly level, PsiCLASS's precision for the GEUVADIS set is 69.3% and 89.2% higher than StringTie's and Scallop plus TACO's, respectively, whereas sensitivity is higher or comparable, by 4.5% and −0.9%, respectively (Fig. 1c). Similarly, advantages of the multi-sample approach are seen for the liver total RNA data set, with more than twofold and threefold increases in precision, albeit at 18.6% and 35.3% lower sensitivity. Even when matched to the other programs' sensitivity settings, PsiCLASS maintains 140% and 89% gains in precision over the StringTie system and Scallop plus TACO, respectively, albeit for this data set Scallop with TACO retains superior sensitivity over the entire parameter range (Fig. 1d). In such cases, the default voting setting may not present the best tradeoff, and as PsiCLASS's precision remains significantly higher than that of its counterparts the user may choose

a different cutoff. Even at the individual sample level, PsiCLASS's performance exceeds that of its competitors', with higher or comparable per sample average measurements in both components (Fig. 1c, d, boxplots).

**Performance evaluation on heterogeneous data collections**. Most RNA-seq analyses are aimed at determining gene expression or splicing differences between two biological conditions. To explore the robustness of PsiCLASS when combining two-condition samples, we applied it and the other methods to RNA-seq samples from hippocampi of normal mice (24 samples) and mice with induced epileptic seizures (20 samples) (ref.[23] and Source Data). The diagrams in Fig. 1e indicate that at the level of individual samples PsiCLASS has higher sensitivity than both StringTie and Scallop, by 26.2% and 8.4% on average, whereas precision exceeds Scallop's by 42.3% and is comparable to StringTie's at 66%. Moreover, after voting PsiCLASS's precision at the level of meta-annotations is 45.7% and 55.0% higher than the other programs', respectively, along with an increase in sensitivity (15.1% and 7.8%), therefore recommending it as the overall best performer. Thus, PsiCLASS can be effectively and

**Table 1 Performance of methods on experiments with small numbers of samples on simulated data**

| Number of samples | Recall PsiCLASS | Recall StringTie | Recall Scallop | Precision PsiCLASS | Precision StringTie | Precision Scallop |
|---|---|---|---|---|---|---|
| 1 | 45.6 | 41.7 | 46.2 | 66.9 | 70.8 | 62.9 |
| 2 | 46.7 | 43.8 | 43.3 | 68.6 | 69.1 | 57.0 |
| 3 | 49.9 | 45.2 | 45.0 | 67.3 | 68.5 | 56.7 |
| 5 | 52.1 | 46.6 | 46.2 | 69.7 | 66.7 | 56.7 |
| 10 | 52.7 | 48.0 | 48.1 | 70.0 | 64.0 | 56.5 |

**Table 2 Performance of methods on experiments with small numbers of samples on real data**

| Sample | Recall PsiCLASS[a] | Recall PsiCLASS[b] | Recall StringTie | Recall Scallop | Precision PsiCLASS[a] | Precision PsiCLASS[b] | Precision StringTie | Precision Scallop |
|---|---|---|---|---|---|---|---|---|
| SRR534319 | 44.2 | 56.7 | 42.0 | 65.7 | 29.5 | 30.1 | 30.5 | 24.5 |
| SRR545695 | 50.4 | 60.3 | 49.4 | 72.7 | 32.5 | 32.4 | 35.7 | 26.6 |
| SRR534291 | 63.3 | 66.1 | 65.5 | 81.6 | 30.8 | 35.0 | 37.1 | 35.1 |

[a]Single samples in one-sample experiments
[b]Individual samples in multi-sample experiments

more reliably used on the aggregate set of samples in a two-condition comparison.

We further explored the applicability of PsiCLASS to more divergent and heterogeneous collections of data. We applied PsiCLASS to single- and multi-tissue collections extracted from the GTEx[24] repository (ten RNA-seq samples for each of six tissues: cortex, frontal cortex, cerebellum, heart, liver, and lung). We then aggregated the results in each collection in three ways, using PsiCLASS voting, TACO, and TACO applied to the samples assembled separately by tissue. In all cases, PsiCLASS with its default voting algorithm had better performance, with both higher sensitivity, and higher precision (Supplementary Fig. 5). Notably, our approach resulted in significantly more accurate transcript sets for all collections, with 11.2–16.8% increase in precision over TACO, and a significant 21.4–47.1% increase over TACO applied to the per tissue assembled collections. As the heterogeneity in the data increases, gains in sensitivity reach diminishing returns, while the increase in precision becomes more pronounced. These simple experiments show promise for extending PsiCLASS's capabilities to more heterogeneous data sets.

**Deconvoluting the effects of algorithmic components**. While the different components of the PsiCLASS algorithm are inter-calibrated to work in harmony with each other, to guide future program development we attempted to deconvolute the effects on performance of the individual algorithmic components. In a first test, we comprehensively evaluated all combinations of individual sample (PsiCLASS, StringTie, and Scallop) and meta-assemblers (voting, ST-merge, and TACO) at multiple levels. Notably, voting significantly increased transcript level precision for all programs, albeit PsiCLASS maintained a clear advantage in all but one case (GEUVADIS-25), where Scallop's performance was comparable (Supplementary Figs. 6 and 7). Conversely, unlike the other two programs, PsiCLASS rendered comparable results when its sample-level transcript sets were merged with the voting, ST-merge and TACO aggregation methods, demonstrating the robustness of its reconstructions (Supplementary Fig. 6). As conventional measurements based on the number of fully reconstructed transcripts only partially describe the accuracy of a data set, we also assessed the accuracy at the feature (exon, intron) and gene level. Indeed, even when the best meta-assembly method (voting) is used for all programs, PsiCLASS captures

significantly more gene and transcript content, with 10–40% more introns and 10–50% more internal exons than StringTie and Scallop, at comparable 95–98% precision (Supplementary Fig. 8). Further, PsiCLASS reported significantly more genes that had at least one fully reconstructed transcript, 7.0–24.7% more than StringTie and 2.9–31.3% more than Scallop. Lastly but notably, PsiCLASS also detected many more genes: 3.9–9.6% more than StringTie and up to a very significant 46.3% more than Scallop, with the largest of these gains observed for the liver data (Supplementary Figs. 9 and 10). These improvements are the direct consequence of the completeness of PsiCLASS gene structure model as captured in the shared subexon graph, an algorithmic feature that particularly favors low expression genes.

**Performance evaluation with small numbers of samples**. We next investigated the utility of the multi-sample approach for small data collections, including single samples. Table 1 shows the results for all methods on sets of 2, 3, 5, and 10 samples from our simulated set, averaged over five independent trials, and for single-sample sets, averaged over all possible 25 trials. The multi-sample approach shows benefits in all cases, most notably increased sensitivity (range 47–53% for PsiCLASS, compared with 42–48% for StringTie, and 43–48% for Scallop with TACO) at comparable or higher precision (range 67–70% for PsiCLASS, compared with 64–71% for StringTie and 57–62% for Scallop).

PsiCLASS is designed to take advantage of cross-sample gene information in a multi-sample experiment. Nevertheless, we also evaluated it and the other programs on isolated single-samples from three previously reported real data sets (Table 2). PolyA-selected RNA-seq paired-end reads from CD20+ (SRR534319, 25 million reads), CD14+ (SRR545695, 39 million reads), and IMR90 (SRR534291, 114 million reads) cell lines were previously used to demonstrate performance improvements in both the StringTie and Scallop publications. When PsiCLASS used solely information from that sample, performance across the three samples was similar between StringTie and PsiCLASS, whereas Scallop had significantly higher sensitivity albeit at slightly lower precision. However, since each data set was part of a multi-sample experiment (SRR534291-SRR534292, SRR545695-SRR545700, and SRR534319-SRR534324), we sought to assess the advantage of the cross-sample information. Even with the small (2–6) number of replicates, when all samples in an experiment were considered PsiCLASS's performance for the

three samples improved (up to 28.3% sensitivity increase over PsiCLASS single-sample, and up to 35.0% over StringTie), especially for the low coverage sample (SRR534319).

Overall, PsiCLASS showed competitive or higher performance on experiments with a small (2–10) number of samples, with benefits most evident for data sets of three or more samples. It also ranked comparable with StringTie and Scallop on single-sample data sets in our control experiments, and similarly to StringTie on the three samples for which the two reference programs were optimized. Therefore, our program can be effectively used for small RNA-seq collections and within the conventional assemble-and-merge approach.

**Performance evaluation on very large data collections**. As the emerging landscape of RNA-seq foresees increasingly larger data sets from large patient cohorts and population variation studies, we aimed to assess the suitability of the multi-sample approach as the data set grows. We evaluated program performance on increasingly larger subsets of RNA-seq samples from the GEU-VADIS population variation project, up to the full set of 667 samples (Fig. 1f). All methods show improvements in sensitivity as the number of samples increases, but while PsiCLASS and Scallop show further slight gains after 20–50 samples, the sensitivity of StringTie drops. Precision drops markedly for both Scallop and StringTie, to less than 35% for 50 samples and below 20% for the full set of samples. In sharp contrast, PsiCLASS's sensitivity and precision remain almost constant with more than ten samples, demonstrating the robustness of this approach. Also, with sensitivity 9–40% higher than StringTie's and precision (>50%) twice as high as that of the other two systems when the data set exceeds 20–50 samples, PsiCLASS is unequivocally the best suited for handling large RNA-seq collections. Lastly, Psi-CLASS took only 9 h with 24 threads to process the 667 samples at a peak memory of 14 GB RAM on an 3.0 GHz Intel "Ivy Bridge" Xeon server, amounting to <1 min per sample. By comparison, StringTie took 34 h to process the samples sequentially, with 24 threads, at a peak memory of 524 MB, and Scallop, which is single-threaded, required 75 h with a peak memory of 5 GB.

## Discussion

Determining the set of expressed genes and transcripts in an RNA-seq experiment is critical for subsequent quantification and differential expression and splicing analyses. The conventional approach to process each sample separately and then merge the sets of transcripts to create a unified set of annotations has limitations, in particular low precision. We present PsiCLASS, a transcript assembler and meta-assembler that simultaneously analyzes all samples in an RNA-seq experiment. Algorithmic underpinnings of PsiCLASS include a global subexon graph, statistical methods for cross-sample feature selection, and dynamic programming optimization and voting algorithms for transcript selection at the level of individual samples and over the entire collection of data, which collectively lead to more complete, more accurate, and more consistent sets of annotations. In particular, our study suggests that voting-based approaches may have advantages over the current assembly-based aggregation methods in some cases, albeit their broader applicability needs to be further investigated.

PsiCLASS is designed to leverage the gene information across samples in a one or two condition experiments, to improve the accuracy of gene and transcript reconstruction for downstream differential gene expression and differential splicing studies, and may not be ideally suited to multi-condition experiments, such as annotation of genes and alternative splicing in large data sets such as GTEx[24] or TCGA[25]. Nevertheless, our limited tests on

collections of samples from 2 to 6 GTEx tissues reflect that Psi-CLASS has better performance than TACO, and therefore show promise for exploring capabilities for handling more heterogeneous data sets in the future.

PsiCLASS had significantly higher precision at similar sensitivity when compared with current best methods, and showed consistently high precision (>50%) when tested on data from a variety of experimental conditions. PsiCLASS is scalable, efficient, and robust with large numbers of RNA-seq samples, thus providing a highly effective paradigm for large-scale analyses of collections of hundreds and thousands of samples. PsiCLASS is available free of charge from https://github.com/splicebox/PsiCLASS.

## Methods

**Algorithm overview**. PsiCLASS builds a global subexon graph of a gene and its splice variants from genome-aligned RNA-seq reads in all input samples. It then traverses the graph to select a subset of the encoded transcripts in each sample. Lastly, it combines the predicted transcript sets across all samples, using a voting procedure to select a final set of meta-annotations.

*Building per sample subexon graphs.* PsiCLASS builds a subexon graph for each sample, then combines graphs across all samples to create a global subexon graph. In each sample, PsiCLASS uses candidate introns extracted from spliced read alignments to divide the genome into regions and subexons. A region denotes a maximal contiguous portion of the genome covered by reads. A subexon is a portion of a region delimited by two consecutive splice junctions and/or the end(s) of the region. A subexon graph has subexons as vertices, and two subexons are connected by an edge if they are adjacent in the same region or connected by an intron. Candidate splice variants are encoded as maximal paths in the subexon graph.

To build the sample-level subexon graph, PsiCLASS clusters read alignments along the genome that are colocated and on the same strand. Introns are extracted from spliced alignments and used to divide the region into subexons. A major confounding factor in determining subexons from RNA-seq data is the presence of intronic unprocessed RNA ('noise'). To differentiate between intronic noise and signal, such as retained introns or alternative 5′ and 3′ gene ends, PsiCLASS assigns each subexon a score that reflects the probability that it is noise. In contrast to current single-sample methods, which simply discard a subexon if it fails sample-wide cutoffs, PsiCLASS then combines sample-level scores across all samples to determine a final label for the subexon and its inclusion in the global subexon graph.

More specifically, PsiCLASS computes the probability that a subexon is due to intronic noise using two models: (i) the exon–intron coverage ratio, and (ii) the intronic read coverage. Let $c_i$ be the average read coverage of (intronic) subexon $i$. In the coverage ratio model, PsiCLASS calculates a score that is equal to the coverage ratio of this subexon versus its flanking subexons: $r_i = \min\left(\frac{c_i}{c_{i-1}}, \frac{c_i}{c_{i+1}}\right)$. The score is fitted to a mixture of two Gamma distributions, one representing signal and one noise: $p(r_i) = \pi\Gamma_{\theta_0,k_0}(r_i) + (1-\pi)\Gamma_{\theta_1,k_1}(r_i)$, where $\pi, (1-\pi)$ are the prior probabilities that an intronic subexon is noise or signal, respectively, and $\theta_0, k_0, \theta_1, k_1$ are the parameters for the Gamma distributions, calculated with an expectation maximization algorithm. With these parameters, PsiCLASS can infer the probability that subexon $i$ is noise according to the Bayes formula:

$$P_R(r_i) = \frac{\pi\Gamma_{\theta_0,k_0}(r_i)}{\pi\Gamma_{\theta_0,k_0}(r_i)+(1-\pi)\Gamma_{\theta_1,k_1}(r_i)}.$$

The coverage ratio model above is insufficient when the overall gene coverage is low. Hence, the second model establishes a similar formula for coverage levels, $P_C(c_i)$, with $\theta'_0, k'_0, \theta'_1, k'_1$ the parameters inferred using coverage. The final per sample subexon score then is $P(i) = \max(P_R(r_i), P_C(c_i))$.

*Building the global subexon graph.* PsiCLASS removes likely artifactual introns and intronic noise subexons by evaluating evidence across all sample, and builds a global subexon graph by combining individual samples' graphs that share at least one intron. *Multi-sample intron selection*: To select a highly accurate set of introns, PsiCLASS assesses each candidate intron's read support across all samples. Assume the experiment contains $M$ samples, and denote each intron by its coordinates in the genome, e.g., $(a, b)$. Let $S(a, b)$ denote the total number of read alignments supporting $(a, b)$ over all samples. Then the total number of alignments supporting its splice sites: $S(a) = \sum_{(a,y)} S(a, y)$, $S(b) = \sum_{(x,b)} S(x, b)$. PsiCLASS keeps intron $(a, b)$ iff: (i) $\frac{S(a,b)}{M} \geq 0.5$, indicating strong read support in one or a few samples, or consistent read support across multiple samples; and (ii) $(a, b)$ appears in at least $M0$ samples, where $M0 = \min\left(\left\lceil\frac{M}{50}\right\rceil\left(\left\lfloor\frac{b-a+1}{100,000}\right\rfloor + 1\right), M\right)$, if $|b - a| \geq 100,000$ (long intron). Condition (ii) is intended to filter out long intron-type alignment artifacts due to gene families and repeats, or from sequencing errors, which can lead to merged genes and transcripts. *Multi-sample subexon selection:* To determine a global set of subexons, PsiCLASS combines the subexon sets of individual samples with some modifications. Where multiple 3′ or 5′-end (i.e., subexons not delimited

by a splice site) candidate subexons occur with the same endpoint and potentially different lengths among the samples, PsiCLASS creates a unique subexon with the median length. Further, to determine intronic subexons, PsiCLASS calculates a final score by combining all sample scores with a Bayesian formula. More specifically, let $\bar{\pi}$ denote the prior probability of intronic noise in the global model, calculated as the average of the mixture coefficients of the samples. Then the subexon score: $P_n(\text{noise}|\text{data}) = \frac{\bar{\pi}P(\text{data}|\text{noise})}{\bar{\pi}P(\text{data}|\text{noise})+(1-\bar{\pi})P(\text{data}|\text{real})}$ reflects the probability that the subexon is noise, where data is the observed information such as the coverage in each sample. We assume the samples are independent, hence $P(\text{data}|\text{noise}) = \prod_{s=1}^{M} G_0^{(s)}(i)$, where $s = 1,..., M$ denotes the sample. Here, $G_0^{(s)}(i)$ is $\Gamma_{\theta_0^{(s)}, k_0^{(s)}}(r_i^{(s)})$ if the ratio model is used for subexon $i$ in sample $s$, and $\Gamma_{\theta_0^{(s)}, k_0^{(s)}}(c_i^{(s)})$ if the *coverage model* is employed. Similarly, $P(\text{data}|\text{real}) = \prod_{s=1}^{M} G_1^{(s)}(i)$. In the end, the subexon is retained if it passes a predefined threshold.

*Transcript selection.* Candidate transcript models are represented as maximal paths in the global subexon graph, from a node with no incoming edges (source) to a node with no outgoing edges (sink). Since the graph generally encodes a much larger number of transcripts than is biologically possible, PsiCLASS identifies and selects a subset of transcripts that can explain all contiguity constraints from spliced reads and paired reads. PsiCLASS first predicts a set of transcripts for each sample, using a graph-based dynamic programming algorithm with the global subexon graph and the sample specific alignment data, then combines the individual samples' transcript sets and selects a subset of meta-annotations by voting.

To predict a set of transcripts for each sample, PsiCLASS employs a SET_COVER framework and dynamic programming algorithms[12], adapted for subexon graphs. Namely, we define a constraint as a cluster of read alignments with the same subexon pattern. Like CLASS2, PsiCLASS uses constraints to decrease the memory usage while preserving the structural and contiguity information contained in the full set of reads. For a given graph $G$, let $C = \{c_1, ..., c_m\}$ denote the set of constraints and $T = \{t_1, ..., t_n\}$ the set of candidate transcripts, encoded in the graph. Given a constraint $c_i$, its abundance $a_i = a(c_i)$ defined as the number of supporting reads (or read pairs) normalized by the number of possible start positions of the reads within the constraint's subexons. To reduce the transcript selection problem to SET_COVER, we view each candidate transcript $t_j$ as the set of constraints that are compatible with its exon-intron structure: $C(t_j) = \{c_1, ...,c_{nj}\}$, where $c_i \sim t_j$. Constraint $c_i$ is compatible with the transcript $t_j$'s exon-intron structure, denoted $c_i \sim t_j$, if its list of subexon intervals is contained in that of the transcript's and all the exon–exon junctions correspond to introns in the transcript.

In the simplest formulation, the goal then is to select a minimal (parsimonious) subset of transcripts that satisfies all constraints. More realistically, to account for the different abundance of constraints, we define a transcript's abundance as the minimum abundance among its set of constraints: $A_j = min\{a_i | c_i \sim t_j\}$. The goal then becomes to determine a subset of transcripts that most closely explain the constraints and their abundance levels. PsiCLASS uses a greedy approximation framework to address this problem, iteratively selecting the transcript that covers the largest number of constraints weighted by the constraints' abundance, then adjusting the constraints' abundance levels before the next iteration:

While ({nondepleted constraints} $\neq \emptyset$):

Choose transcript $t \in T$ that maximizes $|C(t)|\left(1 + \frac{A_t}{A}\right)$

Update the constraints' abundance:

$$x = \min_{c \in C(t)} \{a(c)\}$$

For each $c \in C(t)$:

$$a(c) = a(c) - x$$

if $a(c) \leq 0$, mark constraint $c$ as depleted.

PsiCLASS implements the procedure above in two steps. First, it determines the candidate set of transcripts $T$, using either enumeration (for graphs with <200,000 transcripts) or a variation of the splice-graph dynamic programming algorithm in Song et al.[12] that considers all reads single-end, for fast processing. Once the candidate transcript set $T$ is determined, PsiCLASS applies the greedy SET_COVER approximation algorithm above.

For completeness, we include a brief description of the dynamic programming optimization procedure. The algorithm considers all subpaths $L$, and recursively calculates the maximum number of constraints $f(L)$ for substranscripts starting with subpath $L$: $f(L) = \max_{L'} \{f(L') + c(L, L'),$ if $L'$ exists; $c(L),$ if $L'$ does not exist $\}$, where: (i) $L'$ is a subpath immediately following $L$ so that all constraints compatible with $L$ end before or within $L'$; (ii) $c(L, L')$ is the number of constraints starting in and (partially) compatible with $L$ and $L'$, and compatible with the concatenated subpath $L, L'$; and (iii) $c(L)$ is the number of constraints covered by subtranscript $L$. To take into account the abundance levels in the optimization process, at each sweep of the graph the algorithm excludes subpaths that cover constraints with abundance below a fixed value $x$; hence, the dynamic programming algorithm will return the best transcript with abundance greater than $x$ ($x$-abundance transcript). With this modification, at each graph sweep the selection process selects an

$x$-abundance transcript, starting with $x_0 = 0$ (thus guaranteeing that such a transcript exists), and each selected transcript's abundance value used as lower bound for the selection process at the following step: $0 = x_0 < x_1 < x_2 < \cdots x_m$, until no transcript can be found. The optimal transcript then is among those selected by the sweeps. More details, along with proof of correctness for the algorithm, can be found in Song et al.[12].

*Selecting a global set of transcripts.* PsiCLASS selects a set of meta-annotations from the individual samples' sets of transcripts by voting. Each transcript is assigned a score equal to the transcript's average estimated abundance across the samples (i.e., every sample has one vote, weighted by the transcript's abundance level; default cutoff: 1.0). As different cutoff parameters values might work best for data with specific characteristics, the user can readily adjust or recalibrate the voting parameters postassembly, starting from the already computed sets of transcripts for the individual samples.

**Sequence data.** We generated 25 RNA-seq samples, with ~85 million 100 bp paired-end reads, using the software Polyester[20] with the default gene and transcript distribution models and randomly sampling 10% of the transcripts (at 13,912 genes) from the human GENCODE v.27[26] gene annotations. Reads were aligned to the reference genome hg38 separately with Hisat2 v.2.0.5[21] and STAR v.2.5.3a[22]. Chromosome 2 alignments were extracted and used in the assembly and evaluations. Human liver samples were obtained from the Stanley Medical Research Institute (SMRI) and previously sequenced by Dr. Sabunciyan's lab. All ethical standards set forth by the Johns Hopkins University and local laws were complied with. The liver tissue was collected postmortem by the SMRI after obtaining consent from the family of the deceased. A detailed description of the consent procedure and the collection was published[27]. The project was reviewed by the Johns Hopkins University Institutional Review Board (IRB) and was given exempt status since only postmortem tissue was involved in the research. For sequencing, total RNA was isolated using the Qiagen RNeasy kit and libraries were constructed using the Illumina TruSeq Stranded Total RNA kit for Human/Mouse/Rat following the manufacturers recommended protocol. The resulting stranded, rRNA depleted liver libraries were sequenced on an Illumina HiSeq 2000 instrument. 667 RNA-seq samples from human lymphoblastoid cell lines part of the GEUVADIS population variation project were publicly available from ArrayExpress (accession: E-GEUV-6), and mouse hippocampus RNA-seq data were those reported in ref. [23] and available from GenBank (ProjectID: PRJEB18790). Lastly, RNA-seq samples from six human tissues (frontal cortex, cortex, cerebellum, heart, liver, and lung) used in the heterogeneity study were downloaded from the dbGAP GTEx[24] repository. The lists of samples for each evaluation are included in the Source Data file.

**Performance evaluation.** Once the reads were mapped to the genome, we used StringTie v.1.3.3.b and Scallop v.0.10.2 to assemble them into transcripts, for each individual sample. Transcript sets for all samples in an experiment were then merged with StringTie(ST)-merge and TACO v.0.7.3. For PsiCLASS v.1.0.1, reads were assembled simultaneously across all samples. To evaluate the accuracy of transcript assembly, we employed standard sensitivity (Sn) and precision (Pr) measures and evaluation criteria to assess the accuracy of transcript reconstructions by comparison to a gold reference, namely the set of simulated transcripts and the human GENCODE v.27 and mouse RefSeq gene annotations. A predicted transcript is deemed a true positive (TP) iff its intron chain fully matches that of a gold reference transcript. If N is the number of predicted transcripts, M be the number of ground truth transcripts, then Sn = TP/M and Pr = TP/N [5]. Performance metrics at the transcript, exon, and intron levels were calculated with the tool *grader* included in the PsiCLASS package, and at the gene level with the tool Cuffcompare (v.2.2.1). For gene content analyses, only transcripts that were a full match, contained or splice variant transcript (codes '=', 'c', and 'j') of a reference gene were considered.

**Reporting summary.** Further information on research design is available in the Nature Research Reporting Summary linked to this article.

## Data availability
Raw sequence data for the GEUVADIS project can be obtained from ArrayExpress (Accession:E-GEUV-6), mouse hippocampus data from GenBank (ProjectID: PRJEB18790), and alignments of simulated data from Zenodo (https://doi.org/10.5281/zenodo.1407759). The liver data has been deposited in GenBank under accession PRJNA575230. In addition, the full set of evaluation results represented in Fig. 1, Tables 1 and 2, and Supplementary Figs. 2, 4–9 are provided as a Source Data file, and scripts for performing the evaluation and links to the assembled transcripts can be obtained from the project site in GitHub. All other relevant data are available upon request.

## Code availability
PsiCLASS is available under a GNU GPL license from https://github.com/splicebox/PsiCLASS.

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

## Acknowledgements
Development and evaluations were performed on the Maryland Advanced Research Computing Center (MARCC). Work was supported in part by NSF grants ABI-1356078 and IOS-1339134 to L.F., and by NIH grants R01GM129085 and R01GM124531 to L.F. and Kathleen Burns. S.S. was supported by a grant from the Stanley Medical Research Institute.

## Author contributions
L.F. and L.S. conceived the project, L.S. developed the method, S.S. provided sequence data, and L.S., L.F., G.Y., and S.S. contributed to the evaluation. All authors wrote and approved the paper.

## Competing interests
The authors declare that they have no competing interests.
