## [Peer Review File · Nature Communications]

Editorial Note: This manuscript has been previously reviewed at another journal that is not operating a transparent peer review scheme. This document only contains reviewer comments and rebuttal letters for versions considered at Nature Communications . Mentions of prior referee reports have been redacted.

Reviewers' comments:

Reviewer #1 (Remarks to the Author):

The methodology of this paper is interesting, and authors have done extensive experimental comparisons. The results improve on most datasets, but without addressing the following issues I am still not sure how much the improvement is:

1, In Figure 1 C-E, I did not find the 'additional symbols' that mark sensitivity and precision values for PsiCLASS when tuned to match or approach the sensitivity of its competitors.

2, In the main text of the manuscript, it would be more convincing to report the precision (resp. recall) when recall (resp. precision) has been matched, unless both value outperforms (for examples, line 131 and line 75).

It would be even better if the precision-recall curve can be drawn so that methods can be compared in a broader parameter scale -- that is what I have suggested in my previous comments.

In addition, as the authors have shown that "voting" is better than TACO/StringTie-Merge (Supplementary Figure 6), then placing the comparison between PsiCLASS and StringTie/Scallop followed by voting at the main figure (Figure 1) would make this manuscript stronger. I understand that "voting" has not been reported in published literatures, so this manuscript has no obligation to thoroughly compare with it. Editors can decide whether such comparison needs to be highlighted.

POINT-TO-POINT ANSWERS TO REVIEWERS' COMMENTS:

REVIEWER 1.

Q1. *"The methodology of this paper is interesting, and authors have done extensive experimental comparisons. The results improve on most datasets, but without addressing the following issues I am still not sure how much the improvement is:*

1, In Figure 1 C-E, I did not find the 'additional symbols' that mark sensitivity and precision values for PsiCLASS when tuned to match or approach the sensitivity of its competitors.

2, In the main text of the manuscript, it would be more convincing to report the precision (resp. recall) when recall (resp. precision) has been matched, unless both value outperforms (for examples, line 131 and line 75). It would be even better if the precision-recall curve can be drawn so that methods can be compared in a broader parameter scale -- that is what I have suggested in my previous comments."

Answer: We wish to start by thanking the reviewer for the effort with reviewing the revised version and for the positive remarks on our methodology and on the breadth of our evaluation.

Thanks are also due for noticing the above inconsistency; indeed, the marks were lost during an earlier revision. As requested, we added the symbols and ranges (dotted vertical lines) back to indicate the performance of PsiCLASS if tuned to match the sensitivity of the other methods (**Figure 1**, panels **C** and **D**, GEUVADIS_25 and liver experiments; for all others PsiCLASS outperformed in both recall and precision). Additionally, supporting data for all experiments have been added to the '**Source Data**' document, to allow broader comparisons. Lastly, we amended the text at the points in the manuscript indicated by the reviewer: line 75 (*"iii) 22-140% increase in precision at meta-assembly level over StringTie with ST-merge, and 35-89% over Scallop with TACO, when matched to their sensitivity settings"*), and line 131 (*"Even when matched to the other programs' sensitivity settings, PsiCLASS maintains 140% and 89% gains in precision over the StringTie system and Scallop plus TACO, respectively (Figure 1D)*). Conversely, note that PsiCLASS is more precise or comparable (in one case) than the other methods even in its most relaxed setting (cutoff 0), therefore making the methods *not directly comparable*; even so, PsiCLASS achieves up to 40.3% and 31.3% gains in sensitivity, respectively, over the other methods. We wish to re-emphasize, however, that these marks are not a complete or

accurate measure of the programs' relative performance. All programs operate on a recall/precision curve, choosing defaults that represent their perceived best tradeoff to recommend to their users. In the case of PsiCLASS, that means occasionally trading a small amount of sensitivity for a highly significant increase in precision.

Q2. "In addition, as the authors have shown that "voting" is better than TACO/StringTie-Merge (Supplementary Figure 6), then placing the comparison between PsiCLASS and StringTie/Scallop followed by voting at the main figure (Figure 1) would make this manuscript stronger. I understand that "voting" has not been reported in published literatures, so this manuscript has no obligation to thoroughly compare with it. Editors can decide whether such comparison needs to be highlighted."

Answer: We acknowledge and appreciate the reviewer's views. As we presented in our previous response, including voting as an aggregation method in the comparisons is inappropriate for multiple reasons, including that: i) it is part of *our* algorithm, designed to work specifically with the other components of our method; and, when used with other assemblers, ii) it has mixed benefits and disadvantages, the latter becoming apparent when multiple criteria and accuracy measures are used in the assessment, and iii) its behavior varies widely with different data sets. We kindly point the reader to our previous round of answers, where we addressed these considerations in depth.

REVIEWERS' COMMENTS:

Reviewer #1 (Remarks to the Author):

Regarding Figure 1D:

1, The red square with a plus inside (I assume that it corresponds to Scallop+TACO; it is not clear from the legends on page 17) locates on one side of the dotted line: does it mean that over the entire range (from 0 to 16) the PsiCLASS is not able to match that of Scallop+TACO? I suggest make clear about that.

2, The left panel, Scallop+TACO seems better than PsiCLASS when their precision are matched (the yellow triangle is above the red square with plus). I suggest authors report both sides in the main text (instead of only their performance when sensitivity are matched).

POINT-TO-POINT ANSWERS TO REVIEWERS' COMMENTS:

REVIEWER 1.

Q1. "Regarding Figure 1D:

1, The red square with a plus inside (I assume that it corresponds to Scallop+TACO; it is not clear from the legends on page 17) locates on one side of the dotted line: does it mean that over the entire range (from 0 to 16) the PsiCLASS is not able to match that of Scallop+TACO? I suggest make clear about that."

Answer: Indeed, Scallop with TACO retains higher sensitivity over the entire range (left panel, Figure 1D). We had already noted that in the legend as "tuned to match or approach the sensitivity", and we now explicitly state this in the text (page 4, bottom): "[...] for this data set Scallop with TACO retains superior sensitivity over the entire parameter range".

Q2. "2, The left panel, Scallop+TACO seems better than PsiCLASS when their precision are matched (the yellow triangle is above the red square with plus). I suggest authors report both sides in the main text (instead of only their performance when sensitivity are matched).

Answer: This observation is incorrect, and we believe that it is based on misreading the figure. The left panel refers to sensitivity, which was precisely the data point in question 1 above. Most importantly, for all data sets PsiCLASS exceeds or matches (in one case) the precision of its counterparts over the entire parameter range. Therefore, it cannot be fairly matched for precision to allow a similar comparison.

Further, in all experiments the closest match is achieved trivially for cutoff 0, which can be readily observed from the Source data document and the new figures. To facilitate this, we therefore added the range of values as dotted red lines in Figures 1A and 1E (Figures 1C and 1D already included them).